# A Strategic Process to Manage Collaborative Risks in Supply Chain Networks (SCN) to Improve Resilience and Sustainability

**Marco Nunes [1,*]** , **António Abreu [2,3]** , **Jelena Bagnjuk [4]** , **Edgar Nunes [5]** and **Célia Saraiva [6]**

1   Department of Industrial Engineering, University of Beira Interior, 6201-001 Covilhã, Portugal
2   Department of Mechanical Engineering, Polytechnic Institute of Lisbon, 1959-007 Lisboa, Portugal; ajfa@dem.isel.ipl.pt
3   CTS Uninova, 2829-516 Caparica, Portugal
4   Project Management Department, University Medical Center Eppendorf, Martinistraße 52, 20251 Hamburg, Germany; j.bagnjuk@uke.de
5   Data Scientist—Senior Data Analyst at Deutsche Bank, AG 1 Great Winchester Street, London EC2N 2DB, UK; edgar.c.nunes@gmail.com
6   Department of Informatic Engineering, UTAD-IST, Quinta de Prados, 5000-801 Vila Real, Portugal; celia.saraiva@gmail.com
*   Correspondence: marco.nunes@tetrapak.com or nunesmr@gmail.com

**Abstract:** Resilience and sustainability are two critical factors in supply chain networks (SCNs) to assure business continuity and achieve competitive advantages. Due to the dynamic interconnections between the several parts that comprise a typical SCN such as customers, organizations, sites, departments, geographies, and so on, efficient collaboration between all parts is vital to assure business success, especially in times of uncertainty and unpredictable disruption. Collaborative risks such as poor communication, deficient information exchange, lack of trust, lack or deficient access or reach, just to name a few, that essentially emerge as a result from a shift toward one of the extremes of the collaborative dimension (lack of collaboration or collaborative overload) are very often invisible; however, they are responsible for undesired outcomes such as production defects and delivery delays, just to name a few. In this work, a strategic process to identify and manage collaborative risks in SCNs to help improve resilience and sustainability is proposed. The proposed strategic process analysis contains three key SCN's collaborative dimensions ((1) network access or reach, (2) trust, and (3) communication) applying graph centrality metrics, looking for emergent collaborative risks in a quantitative way that potentially may threaten an organization's efficiency and performance, and thus negatively impact resilience and sustainability. A case study conducted in the middle of the COVID-19 pandemic is illustrated to describe how organization benefit regarding the timely and quantitative identification of potential behavioral patterns that lead to one of the collaborative extremes. The results show that the application proposed strategic process is very successful in ensuring sustainability improving resilience of SCNs.

**Keywords:** supply chain networks; collaborative risks; risk management; network analysis; resilience; sustainability

## 1. Introduction

In today's business landscape characterized by turmoil, unpredictable and disruptive events and growing customer demand and scrutiny, if organizations want to assure business continuity or even just survive, they must craft strategies that enable them to achieve sustainable competitive advantages while assuring that they remain or become more resilient to face unforeseen disruptive events that tend to take place more often and with higher impacts for organizations [1–3]. For example, the disruptions caused by the SARS crisis in 2003, the tsunami disaster of 2004, the Global Financial Crisis in 2008, the outbreak

of the COVID-19 pandemic, or the Russia's invasion in Ukraine in early March are just some examples of huge and unpredictable disruptive events. Particularly, the still ongoing COVID-19 pandemics caused an unparalleled negative impact in terms of business-chain length and geographical reach, disrupting organizations like never before, with severe consequences for business and society that are still not entirely understood [4,5]. For example, the COVID-19 outbreak has been particularly hard in the global production system impacting over 75% of the world's global manufacturing outputs This happened essentially due to forced factory shutdowns while huge demand for essential goods surges, stockpiling and panic buying. Moreover, the shift in consumer preferences (e.g., online over physical) has raised new and unprecedented questions regarding the level of resilience of global value chains and the overall approach to supply chain management [4].

If organizations want to be more prepared to deal with similar even harder future shocks, they must work together with governments, institutes, suppliers, and even competitors, to develop new forms of collaboration across companies and industries to ensure business continuity and the achievement of sustainable competitive advantages, while protecting employees and improving future supply systems resilience [1,4–6]. In fact, several research studies show that efficient collaboration within and between organizations is one of the most critical elements to achieve success, especially in strongly unpredictable times [7–9]. Such results show that organizations should invest more toward the improvement of their collaborative dimensions, namely their respective processes. Research also shows that regardless of the business area or industry type, if organizations efficiently work in networks of collaboration, they will largely increase their success [7–11]. Furthermore, research shows that the ability to efficiently work in collaborative networks is twice a predictor of success than individual know how and expertise [8,9].

In supply chain networks (SCN), the optimization processes are traditionally based on cost-competitiveness reasons, and very little is based on the improvement of the existing collaborative dimensions [4,12–14]. However, the COVID-19 pandemic has proved that companies also need to improve their collaborative processes and procedures mitigating risks that may threat the collaborative dimension [2–4,12–15]. This means that SCNs must simultaneously increase their use of local suppliers and manufacturing capacities, diversify their supply base to protect supply, create transparency, optimize production and distribution capacity, assess realistic final-customer demand, estimate available inventory, identify secure logistics capacity, tailoring of manufacturing and supply systems to changing consumer behavior, apply agile approaches and advanced technology, reduce complexity, and improve and/or redefine collaborative relationships to efficiently manage potential emerging collaborative risks. Several research studies show that managing collaborative risks is done more efficiently when performed under a data-driven approach [1,4,15,16]. Given that behind a particular process, procedure, or even a machine the likelihood of having a human being is very high, this implies the application of data science concepts (essentially data analytics) to manage collaborative relationships aiming to improve SCN's resilience and sustainability. Although the use of data analytic models is not something new, due to the easy access to latest technology and tools that enable us to record and process tons of data in a blink of an eye, it has been growing in popularity across almost every business area [1,2,6,16]. Their popularity span can be seen in the different business areas, such as from predicting of tool wear in advanced manufacturing industries [3], to the identification of project critical success factors by analyzing dynamic behavioral patterns [1,17], and to the improvement of additive manufacturing by application of advanced data analytics [2].

The concept of resilience combines both agility and robustness and represents the ability of a system to efficiently deal with change [10,18]. In other words, resilience translates how good or how bad a system can recover from a disruptive event. This concept is transversal to several areas of an SCN that may range from the physical arrangements of the different manufacturing sites and departments to the collaborative dimension of such SCN elements (people, departments, groups, organizations), just to name a few [12]. In the collaborative dimensions, research shows that efficient collaboration across the whole SCN

(which essentially implies the efficient management of potential emerging collaborative risks) can help to reinforce the whole supply chain ecosystem resilience [19]. However, achieving an effective and efficient collaborative level within and between organizations is not an easy task for most organizations, especially in the actual days which are characterized by huge and complex chain interdependency, where very likely any emergent disruption is immediately amplified, and its recovery is more prolonged [4]. Not only is such collaborative level hard to achieve, but also several research studies show that there is still a lack of efficient and proper models to manage organizational collaborative initiatives [1,5,19].

In this work, a strategic process to manage collaborative risks in SCNs to improve their resilience and sustainability is proposed. The strategic process built based on three scientific pillars (Figure 1) will analyze how dynamic patterns of behavior that emerge and evolve across time can potentially compromise the functioning on SCNs.

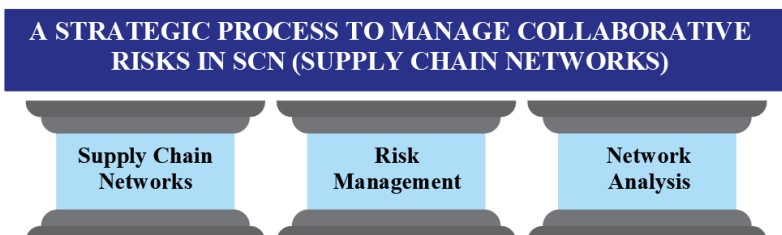

**Figure 1.** The three scientific pillars that support the development of the proposed strategic process.

The proposed strategic process aims to answer the following research quarrion: how do organizations timely and efficiently predict and avoid being disrupted due to the heading toward one of the two collaborative extremes ((1) lack of collaboration or (2) collaborative overload) and their known negative effects in achieving outputs and outcomes, while simultaneously strengthening resilience and sustainability? To answer the research question, the proposed strategic process will analyze three key SCNs collaborative dimensions that, according to much research, are key to improving SCNs resilience and sustainability [2,4,12,15]. They are as follows: (1) network access or reach, which concern the risks related to the lack or limited of access to certain key elements within a manufacturing network; (2) trust, which concerns the risks related to the lack or poor trust level within a manufacturing network; and (3) communication, which concerns the risks of having poor or deficient communication within a manufacturing network. The results obtained through the application of the proposed strategic process can be used by organizations as collaborative key performance indicators of SCNs resilience and sustainability shedding light on how organizations can become more agile (a nimbler way of working by simplifying processes and procedures) and flexible (assuring efficient task redundancy) helping them to achieve sustainable competitive advantages. The research presented in this work is critical to organizations because it contributes to close a gap that concerns the management of collaboration in organizations, by introducing a more data-driven approach in the process of identification, analysis, and treatment of potential collaborative risks that may emerge and evolve in SCNs.

## 2. Literature Review

### 2.1. Supply Chain Networks

Supply chain networks (SCNs) can be defined as a group of different facilities owned by different organizations, geographically dispersed with different organizational management and structures styles, with a high level of interdependence and connectivity between participating organizations [2,13,14]. SCNs originate from manufacturing networks, which in turn originate from operations management of a single facility [2,13,14,20].

Because both terms (manufacturing networks and supply chain networks) are often interchangeable (however different), it is critical to understand the key differences between them.

A manufacturing network is owned by one single organization, whereas a SCN is comprised by a complex number of different partners from numerous organizations, spanning several countries and even continents that work together sharing expertise and competencies in one or several areas to produce and deliver an end customer product [2,13,14,20].

Manufacturing networks can be seen as internal networks to an organization which are managed through an operations management approach focusing on the management of nodes (different facilities, sites, departments) [14]. SCN can be classified as external networks to an organization (links between different organizations) which are managed through a logistic management perspective focusing on the links (the connection channels between the different organizations) instead of the nodes [14].

Manufacturing networks differ from SCNs in three major factors: (1) number of sites, (2) number of organizations, and (3) ownership of the different facilities [13,14]. In Table 1 are illustrated the major differences between manufacturing networks and supply chain networks using the two major factors (number of organizations, and number of sites) as well as the respective physical configuration and the typical coordination style. The third factor (ownership of the different facilities) is always implicitly associated to an SCN.

**Table 1.** Manufacturing networks and SCNs differences.

| **(1)　Number of Organizations** | **(2)　Number of Sites** | |
| --- | --- | --- |
| | **Single Site** | **Multiple Site** |
| Single organization (Manufacturing Network) | • Plant <br> (Single organization/site) <br> • Utilize | • Intra-firm network <br> (Single-organization, multi-site) <br> • Optimize |
| Multi-organization (Supply-chain Network) | • Supply-chain network <br> (multi-organization, single site) <br> • Synchronize | • Inter-firm network <br> (multi-organization, multi-site) <br> • Harmonize |

According to much of the literature, the research on supply chain and manufacturing networks we can divided them in two areas [12–14]: (1) the first area regards to the physical configuration (number of sites and number of organizations as illustrated in Table 1), and the second (2) concerns the different coordination styles (management of inter and intra-collaboration that range from utilization, optimization, synchronization, and harmonization as illustrated in Table 1).

According to Table 1 a SCN is considered a multi-organization network and can be divided into single-site (usually called as supply-chain network) and multiple-site (usually called as inter-firm network), while a manufacturing network is considered a single-organization network that can be divided into single site (usually called a plant) and multiple-site (usually called intra-firm network).

For each one of the four types regarding the number of sites illustrated in Table 1, a specific (more often seen) management style is adopted.

In the single-organization single-site the usual management approach adopted is essentially the utilization of existing resources, which means to efficiently use all available resources within a given site or plant.

In the single-organization multiple-site, the usual management approach is optimization. In such cases, where multiple sites that work together in sequence or parallel through

vertical or horizontal network structures, optimization (optimal allocation of products and volumes to plants, production, and distribution of products) is the most popular management style.

In the multi-organization single-site, network synchronization is the most popular management approach. Here the focus is almost always on the internal (within the given site) operations only. In such cases, the coordination and the management of collaboration is of critical importance.

In the most complex arrangement classified as interfirm networks (multi-organization and multiple site), harmonization is the most popular management approach. Here, the issues involved are "beyond" utilization, optimization, and synchronization, and therefore the level of management is "reduced" to harmonization, which essentially consists of dealing with the coordination and management of the collaboration of the hyper-connections between different facilities, people, finance, and systems in a high-level approach. Here, the focus is essentially put upon the management of collaboration and feasibility of the whole network.

An SCN is a global network that comprises billions of interconnected interactions between people, places, and production processes around the planet, used to efficiently deliver products and services from raw materials to end customers through an engineered flow of information, physical distribution, and cash [21–23]. However, this does not happen without facing problems or challenges that may affect a critical SCN´s aspect and resilience.

In SCN, resilience can be defined as a set of organizational capabilities to face immediate and unexpected changes in the environment with proactive and reactive actions to anticipate, adapt, respond, recover, and learn from predicted and unpredicted disruptive events [22]. More concretely, resilience has to do with how organization deals with problems or challenges, such as for example an earthquake, an economic recession, a corporate scandal, a bankruptcy, economic sanctions, inefficient collaboration, leak of classified information, among many others. Resilience has two major aspects which can be accessed in terms of time and/or money that is needed to restore initial conditions [22,23]. They are the (1) capacity to resist a disruptive force or event (which essentially focuses on avoidance and containment of a potential disruptive event), and the (2) capacity to recover to the initial state (which essentially focuses on stabilization on the consequences of the disruptive event and return to an initial state before the disruptive event).

Businesses are going about an incredible transformation as never seen before in history. Due to the current technology and connectivity's capabilities, SCNs are moving from the linear single cell organisms' chain to hyperconnected nonlinear complex networks that look and act like complex multi-cellular organisms interacting with every aspect of human activity in a complex adaptive system composed of multiple agents constantly adapting to each other's behavior. However, such transformation may bring new management-related problems or challenges, especially in a time where the likelihood of ripple disruptive events that potentially threatens SCN´s resilience is high as never before, [24]. Although there are some efficient strategies to deal with SCNs issues, such as task redundancy, good will, robustness, and efficient contingency planning, just to name a few, these seem to be far from efficient to deal with collaborative risks particularly [22,23]. Although SCNs are becoming flat-hierarchical networked structures with different arrangement flows, their nonlinear high-complexity interdependent connections may lead to the emergence of collaborative risks, such as organizational silos, information bottlenecks, and poor or deficient information exchange between involved parts, just to name a few [22,23]. In this work, the proposed strategic process aims to identify, analyze, and suggest treatment strategies of potential emergent collaborative risks that may emerge as SCNs involved parts dynamically interact in the most complex arrangement in SCN characterized by multi-organizations and multiple sites as illustrated in Table 1.

*2.2. Risk Management*

Risk management is comprised of a set of coordinated activities that should be supported and transversally incentivized across the organizational structure and dimensions including governance, operations, strategy, culture, and so on, to direct and control an organization regarding risk comprising activities, such as risk identification, risk analysis, risk mitigation, and risk treatment (decision making) [25].

Often, risk is usually associated with a given threat; however, risk has two dimensions [1,25]. They are the following: (1) threats, which are events that if occurs will negatively impact organizational goals and objectives; and (2) opportunities, which are events is that occur they will positively impact organizational goals and objectives. Risks can be classified in four different types [1,19]: (1) event risks (risks related to some event that has not taken place yet but would impact organizational objectives if it did); (2) variability risks (risks related to a possible known number of events but nobody knows exactly which one will occur); (3) ambiguity risks (risks that emerge from the lack of knowledge of how things work or should be done); and (4) emergent risks (risks that are outside of a human´s mindset and whose occurrence is therefore practically impossible to predict). According to much of the literature, the above-mentioned risks can be managed by the application of lessons learned and simulations, and the well-known risk management tools and techniques as proposed by the ISO 31000 [1,9,26–28]. However, another type of risk that many times is neglected by organizations may emerge in SCNs (but not only) is known as collaborative risks [1,28–30]. In fact, this type of risk may potentially exist whenever there are people working together (collaboration, coordinating or cooperating) to achieve strategic goals and objectives. Such risks usually emerge resulting from a non-efficient collaboration state between the people that work together to deliver on objectives and may include risk of assigning tasks to partners, risk of critical enterprises, risk of information sharing, risk of access to other parties, among many others.

Due to the increase in uncertainty and disruptive environments worldwide, the importance of risk management has been increasing over the past years. Efficient risk management in SCNs will strongly shape process landscapes across the entire value chain, enabling organizations to quickly develop dedicated plans which can be immediately kicked into action [4].

One of the most popular standards on risk management is suggested by the International Organization for Standardization (ISO) in the ISO standard 31000:2018—risk management, guideline's standard [26]. This ISO 31000:2018 standard has such popularity due to its capacity of being efficiently adjusted to any type of industry or business type, size, complexity and consists in essentially six steps [26]. Step 1: establishing scope, context, and criteria, which is comprised of the definition of the scope of the risk management activities, including the internal and external context, and the amount and type of risk that an organization is willing to take, relative to their objectives. Step 2: risk identification, which includes finding, recognizing and describing risks that hinder or enhance an organization to achieve its objectives. Step 3: risk analysis, which is the process of understanding the nature of risks in different dimensions, including uncertainties, risk sources, consequences, likelihood, events, scenarios, controls, and their effectiveness. Step 4: risk evaluation, which is the process of comparing the results of the risk analysis with the previously established organizational risk criteria to uncover where additional action is still required. Step 5: risk treatment, which encompasses the specification of how to choose treatment options to be implemented. Step 6: record and report previous steps, which is the process of continuously monitoring and reviewing the identified risks evolution across time, and the efficacy of applied control or corrective measures.

In this work, the functioning principle of the proposed strategic process to manage collaborative risks in SCNs is partly inspired in the ISO standard 31000:2018—risk management, guidelines standard [26]. More concretely, the strategic process steps that include target preparation, data collection, data preparation, data analysis, output and results anal-

ysis are based on the official steps of the risk mentioned management standard from the ISO. The detailed strategic process steps are illustrated in Section 2.4, Collaborative Risks.

### 2.3. Network Analysis

Network analysis (NA) studies and analysis networked structures applying several theory-based metrics graphs have been developed, which enables us to explain how social structures emerge and evolve, and eventually disappear across time, and how they impact the environment where they exist [28–30]. The analysis of networks involving dynamic entities relationships such as people, systems, or mechanisms is usually known as social network analysis (SNA) [1,25].

The application of SNA has become very popular and spans across several areas such as leadership [31,32], behavioral sciences [32], organizational performance [33], stakeholder analysis [34], public engineering projects [35], just to name a few.

SNA can be classified as a set of specific links among a given set of entities (people, groups, organizations, and so on), where such links are used to understand social behavior of the involved entities [36]. SNA is critical in identifying social capital challenges and has been continuously integrated into traditional organizational risk management processes, essentially to provide a more data-driven approach in decision making [37].

In organizations (regardless of type of industry or business and size), SNA can be applied to analyze several organizational dimensions, such as collective and individual performance, employee retention and turnover, culture, social cohesion, innovation, information diffusion, collaborative risks, values, ethics, behavior, wellness, satisfaction, fraud, and many others [1,33]. Much research shows that dynamic interactions between elements of a given social network (the mix of informal and formal collaborative patterns) are complex by nature and cannot be understood and explained by the application of traditional social theory methods, instead by methods that are based in sociology, where the individual´s social context in the process of decision making (making choices) is taken into consideration [38–40]. Furthermore, research suggest that the application of SNA tools and techniques in organizations is the only effective way to uncover hidden collaborative patterns within the mix of formal and informal networks [40].

Although the benefits of applying SNA in organization is considerable, the application of SNA in organizations is still in a very initial phase [41].

For example, SNA can be applied to identify key collaborative networks such as (1) advice network (uncovers people to whom others go to in order to have their job done), (2) trust network (uncovers people who share project-related information), and (3) communication network (uncovers people who talk about project-related matters) that still today play a fundamental role in organizational performance and innovation [33,42,43].

Through the application of SNA centrality metrics, key roles within a social network have been identified regardless of an organization type, size, or objective [31,33,40]. They are central connectors (people to whom others turn to gain support or advice to execute their tasks or activities), boundary spanners (people that connect different organizations and departments), information brokers (people that connect different areas within an organization), peripheral experts (people that may be subject matter experts or miss integrated in the organizational social network) and energizers (people who energize others around them with positive energy).

Another interesting study applying SNA is illustrated by [7] where SNA was applied to uncover and efficiently re-connect two critical organizational areas (operational and entrepreneur), in an environment that he called as adaptative space. This enables us to create and explore new ideas in a more agile way.

Much research in the field of SNA suggests that SNA centrality metrics, such as in, out and total-degree (indicator of a network's activity potential), closeness (indicator of the independence potential of a network) or betweenness (indicator of control and communication between two different groups), just to name a few, enable to us obtain

better insights into a given social network, enabling thus the application of more actionable and practical corrective or supportive measures.

Research shows that centrality is strongly correlated with informal power within a collaborative network, that strongly influences coordination and decision making [38,40,44]. Finally, the application of SNA centrality metrics in organizations is often used to efficiently measure informal importance, influence, prestige, prominence, and control, of entities within a social network [1,32,33,40–44].

In this work, SNA centrality metrics such as in-degree, total-degree, centralization-degree, average in-degree, density, and reciprocity, will be applied to quantitatively measure potential emergent collaborative risks in SCNs. These metrics that belong to the centrality graph-based metrics have been chosen as suggested by several research studies as being the ones that generate most valuable, insightful, and actionable outputs when analysing dynamic interaction between entities within a particular environment [1,27,32–34].

### 2.4. Collaborative Risks

Collaborative risks are risks that essentially emerge from deficient collaboration between the different organizational departments of a given organization or between different department of different organizations [8,9,32].

More concretely, the emergence of collaborative risks result from a deviation from the collaborative center heading toward one of the two collaborative extremes which are known as (1) lack or inexistence of collaboration within a given organizational social network, and (2) collaborative overload within a given organizational social network [19,29,40]. Based on the researched literature [1,19,29,32,40], we illustrate in Figure 2 the two collaborative extremes and the respective potential collaborative risks, and the collaborative center.

The two collaborative extremes represented in Figure 2 that may emerge within a given organizational social network result from the emergence and evolution of the mix of formal and informal dynamic interactions (relationships) between people across a period function of the work specificities interactions. In Figure 2, the black circles in each one of the boxes represent people of a given organization, while the red and green arrows represent the given interactions (relationship) between them. According to research, such interactions may represent trust, like, dislike, advice, and so on [1,29,32]. In the left side of Figure 2 is illustrated one of the collaborative extremes which is characterized by two different states, however with the same index value of zero (0). State 1 (*a1* in Figure 2), lack or inexistence of network collaboration characterizes a given organization social networks where there is no interaction between any of the element of a social network. State 2 (*a2* in Figure 2), total network collaborative overload (also known as chaos) characterizes a given organization social networks where there is an excess of interactions between any of the element of a social network, which may lead to the emergence of chaos. In the right side of Figure 2 (*c* in Figure 2) is illustrated the other collaborative extreme which is characterized by the single state of index value of one (1). This extreme represents when in an organizational social network, there is an absolute dependence or collaborative dynamics in one way only (one person only), leading simultaneously to a total individual collaborative overload, and a total lack or inexistence of network collaboration.

These two extremes are characterizing according to their numerical outcomes (either 1 or 0) that result from the application of social network analysis centrality metrics, while the rest of the spectrum varies between 0 and 1 (*b* in Figure 2). In fact, it is not possible to determine within the spectrum between 0 and 1 with an exact numerical value any type of ideal collaboration (interaction) that leads organizations to success. This occurs due to very different human behavioral and organizational factors, such as how much can different people deal a given workload level before breaking down or burning out (what for some a given workload is acceptable, for others may be impossible to execute), the specificity of tasks and activities of a particular job (military, call-center, innovation departments, and so on), and still how a particular organizational social network is formally organized to execute tasks and work activities (the formal chart).

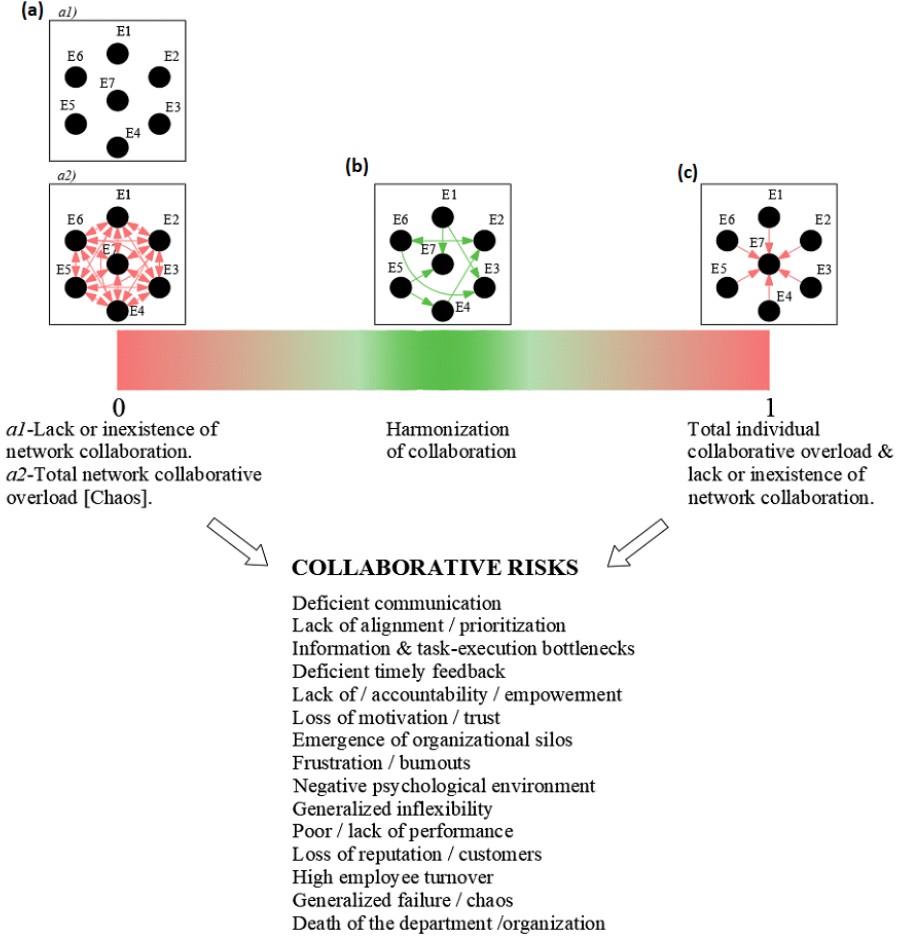

**Figure 2.** Collaborative extremes heat map and respective risks—(**a**) lack or inexistence of collaboration or total network collaborative overload, (**b**) harmonization of collaboration, (**c**) total individual collaborative overload & lack or inexistence of network collaboration.

However, according to several research studies [12,19,30,33,40], if collaboration within and between organizations is not efficiently managed (regarding the emergence of collaborative risks), it is very likely that, sooner or later, the organizational collaboration dynamics will end up falling into one of the two mentioned extremes (1 or 0), and these put at high risk the achievement of an organization's goals and objectives. To avoid falling into one of the two mentioned extremes, the proposed strategic process acts preventively, characterizing how collaboration is evolving at a given point within an organization by calculating a numerical value between 1 and 0, which helps us to predict towards which extreme the collaboration is heading. By acting so, the proposed strategic process contributes to increase the resilience of a SCN social network preventing that collaboration evolves towards one of the mentioned extremes which very likely may lead to collaborative risks, such as deficient or poor communication, lack of alignment and many others as illustrated in Figure 2. The strategic process acts on one side identifying where the necessary and sufficient connections between entities of a network must be rebuilt in case the observed organizational behavioral trend heads to a unique individual collaborative overload (Index value 1 in Figure 2), and on the other side by identifying where must the connections be recalibrated (rearranged) in case the observed organizational behavioral trend heads to a lack or inexistence of collaboration or total collaborative overload (Index value 0 in Figure 2). As a direct consequence of balancing the sufficient and necessary connections between entities in an SCN, the strategic process contributes to increase the sustainability of the SCN social network, which leads to save extra resources (people, time, energy, and

so on) required to accomplish or re-accomplish tasks and activities in case the collaboration would fall into one of the mentioned extremes (0 or 1).

Much research argues that efficient collaboration within and between organizations (regardless of type, size, and business area) is by far a higher predictor of success than individual knowledge and experience [1,8]. Moreover, much research points out that there is a lack of efficient models to manage the collaborative dimension, and as a consequence, organizations suffer major losses due to poor or lack of timely management of collaborative activities within and between organizations [11,15,40]. The strategic process proposed in this work is to be seen as a contribution to cover the gap of models to efficiently manage collaborative extremes in organizations as proposed by several research studies [1,15,40], concretely aiming the management of collaborative risk in SCNs.

### 3. Materials and Methods

*3.1. Development of the Strategic Process Proposed in This Work*

The proposed strategic process in this work to identify collaborative risks in SCNs is supported in three main pillars: (1) supply chain networks, (2) risk management, and (3) social network analysis. Their individual contribution to the development of the strategic process is illustrated in Table 2.

**Table 2.** Contributions of each pillar to the development of the model.

| | |
|---|---|
| Supply chain networks | This pillar contributes to the strategic process with the definitions, terminology, characteristics, and structure of SCN. It includes the specificities of a manufacturing network (characterization regarding the number of organizations and number of sites) as well as the major issues associated to operational environment, such as the most appropriated management approach to manage collaborative risks, and the physical design of a manufacturing network. |
| Risk Management | This pillar contributes to the strategic process with the "best practice approach" steps to perform risk management in organization. It has its foundations in the ISO 31000 risk management standard [26]. This standard provides the principal steps that an organization should adopt to efficiently perform the activities that comprises the process of risk management which includes major activities such as risk identification, risk analysis, risk mitigation, and risk treatment. |
| Social Network Analysis | This pillar contributes to the strategic process with the mathematical approach to quantitatively measure collaborative risks in SCNs. Essentially this pillar provides the metrics, which are based on graph theory [32], to uncover hidden behavioral patterns of collaboration that emerge and evolve across a given period within an organizational social network. The metrics provided by this pillar are essentially SNA centrality metrics, which according to several research studies [1,19,32,38,39] are the ones that better translates and correlates the myriad of relationships within a given organizational social network with outputs and outcomes. |

The proposed strategic process will analyze three majors different—but inter-related—dimensions that co-exist within an organizational social network. According to several research studies, three dimensions are critical within a given organization social network which almost completely dictate whether an organization will fail or succeed regardless of the type of industry organizational structure [1,5,8,9,38–44]. They are (1) access or reach, (2) trust, and (3) communication. In Table 3 are illustrated the objectives of analyzing the three collaborative dimensions in the context of the proposed strategic process to manage collaborative risks in SCNs.

**Table 3.** Detailed objectives of the analysis of each dimension of collaboration.

| Dimensions | Description and Objectives |
|---|---|
| (D1) Network Access (more access to) or Reach | Network access or reach respects the capacity that one entity within a social network must reach or access all the other entities within the same social network. In one side, if an entity of a social network has access to all the other entities of that same network it may be beneficial to the individual and the group because it may mean fast access to vital information to perform work efficiently and general well-being [1,8]. Simultaneously, it may also represent disorder and confusion because it may mean the existence of chaos and too much redundancy regarding work performance and well-being [1,8]. On the other side, poor or no access of an entity to another entity within the same social network may represent a threat to work performance and general well-being because it hinders the interaction level and thus information exchange, creation of bounds fostered by the mix of formal and informal relationships, just to name a few, resulting into a strong siloed organizational structure [1]. This may occur due to several factors, such as lack of time of some entities of a social network, cultural differences fear, lack of psychological safety and so on [45]. By mapping the network access (more access to) or reach, the strategic process will identify which elements are becoming critical (extremely central) for the information exchange process within a manufacturing network. This will enable to correlate arrangements visible in the mapping of this network with organizational outcomes and outputs. |
| (D2) Network Trust | The trust level within a given social network can characterize how efficient and with what frequency does information flows across the whole network [1]. Being trust a fundamental factor to efficiently spark organizational interactions [1,8], such as communication, learning, teaching, validation, problem solving, decision making, just to name a few, it is critical to ascertain the level of trust within a social network. Mapping the trust network of a manufacturing social network enables us to identify critical partners or subnetworks, whereby trust and support (translated into professional and personal) is established. Aspects, such as intensity, frequency, confidence, empowerment, and reliability, are entitled to be analyzed in the trust network [1,8–11]. |
| (D3) Network Communication | The mapping of the communication network in a manufacturing social network enables to analyze aspects related to how effective, efficient, and centralized (or de-centralized) communication takes place between the different elements that collaborate to accomplish organizational tasks or activities [1]. Aspects such as frequency, intensity, and broadness are analyzed [9]. In this case, only the communication channels under a given frequency between the different elements that comprise a manufacturing network will be analyzed. By doing so, one can identify communication bottlenecks, and the emerging of organizational silos. Furthermore, one can identify given elements within a manufacturing network that due to their structural position within the network, may be more susceptible to face damages (burnouts, frustration, among others), which may end by affecting the whole manufacturing social network [1,8–11]. |

As it can be seen in the table above, each one of the three critical dimensions of a given organizational social network has its own specificities, however deeply inter-related within each other.

### 3.2. Implementation to the Strategic Process Proposed in This Work

The implementation of the proposed strategic process in this work to manage collaborative risks in SCNs is illustrated in Figure 3.

In Figure 3 is illustrated the implementation framework of the strategic process proposed in this work. The framework essentially describes the collection and treatment process of collaborative data across time, needed to fuel the proposed strategic process. First, a beginning and an end should be defined to fence the analysis. Then, the collection points (*CP*) between the start and the end of the analysis should be agreed upon and defined by the organization and network analyst. Second, in each one of the CP points, collaborative data that respects a given period "Period *n*" will be collected. The periods of time *n* are customizable are defined by a t variable that represents time, which may be represent hours, days, months or even years as it is illustrated in the legend of Figure 3. Finally, collaborative data will then be processed and displayed in a directional and nondirectional graph-networked form to be further analyzed as it is illustrated in all the four boxes essentially displaying hatched circles (representing people) and the respective links between them (representing relationships between people). As mentioned before, the strategic process developed in this work is inspired in the risk management standard

developed by the ISO institute. The relationships between the standard and the proposed strategic process are illustrated in Table 4.

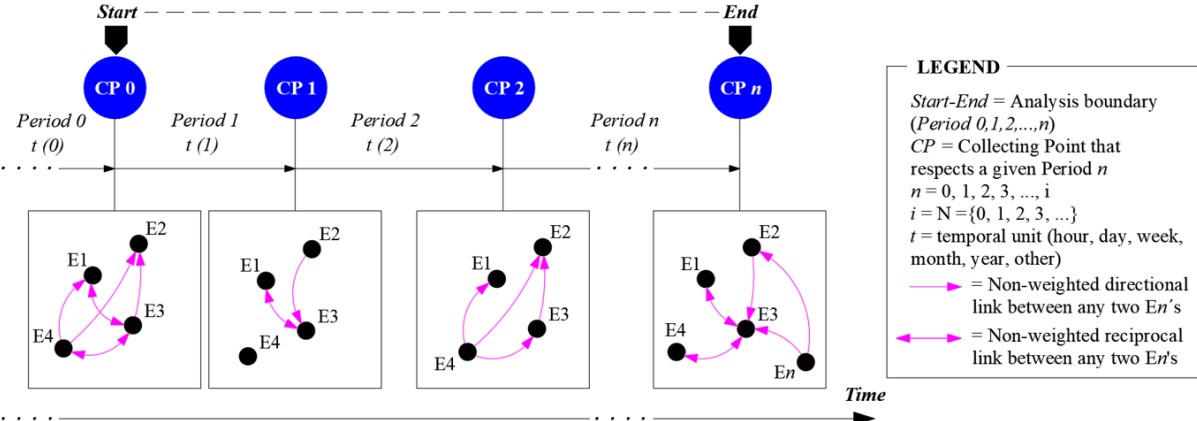

**Figure 3.** Implementation framework of the proposed strategic process.

**Table 4.** Proposed strategic process functioning process.

| Risk Management Steps according to the ISO 31000:2018 | Proposed Strategic Process Equivalent Steps |
| --- | --- |
| Step 1: establish scope, context and criteria | Step 1, data funnel: Define target and collect relational information to map the three key manufacturing network's collaborative dimensions (network access or reach, trust, and communication) |
| Step 2: risk identification | Step 2, data preparation: apply metrics to collected data, to quantitatively identify potential collaborative behavioral patterns risks |
| Step 3: risk analysis | Step 3, data processing: analyze the results and correlated them with work performance (outputs and outcomes) |
| Step 4: risk evaluation | Step 4, analysis of results and decision-making. evaluate the impact of identified collaborative risks in work related matters and decide the implementation of corrective or supportive measures |
| Step 5: risk treatment | Step 5, apply changes and follow up: implement decided corrective or supportive measures |
| Step 6: monitoring and reviewing | Step 6, monitoring and feedback: continuously monitor implemented measures and assess their effectiveness. Generate lessons learned |

As it can be seen in Table 4, the proposed strategic process frames the ISO standard 31000:2018—Risk management, Guidelines standard steps throughout the identification of potential collaborative risks in SCNs to increase resilience and sustainability. The detailed procedure that takes place in each one of the CP points is illustrated in Figure 4.

In Figure 4 is illustrated the methodology of the proposed strategic process to manage collaborative risks in SCNs. Figure 4 details all the necessary steps to efficiently perform the analysis regarding the identification of possible collaborative risks. The complete process has six steps ((1) Data Funnel, (2) data preparation, (3) data processing, (4) analysis of results and decision making, (5) apply changes and follow up, and finally (6) monitoring and feedback).

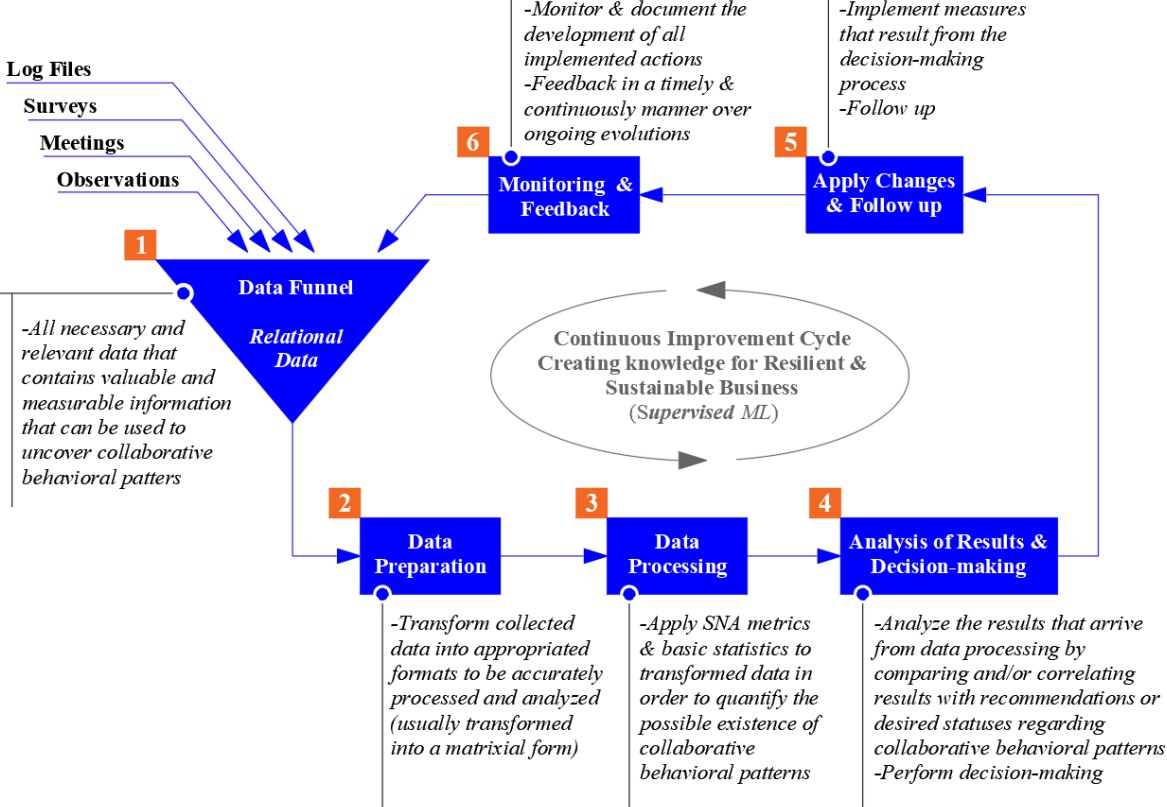

**Figure 4.** Methodology of the proposed strategic process to manage collaborative risks in SCNs.

In step 1 (data funnel) are collected all the necessary and relevant relational data (data from which collaborative behavioral patterns may be identified). These data may be collected in organizational log files such as emails, chat channels conversations, and so on, surveys addressed to the employees of an organization (also known as strategic questionnaires), group meetings where people are questioned regarding collaborative initiatives, and observations on site where a network analyst observes the different behavioral patterns of the different employees of an organization. For example, if the objective is to map interactions between different entities within a given social network in terms of who provides help or advice regarding work matter related, exchanged emails between such entities can be checked out to search for who sent emails to whom looking for help or advice, in case the collection data method is to log files. If the data collection method used is a survey, then questions such as *whom do you turn to in order to gain support or advice when you have a problem with a particular work task or activity* or *whom do you provide help or support regarding a particular work task or activity?* If the collection data method used is observation, then the network analyst should be where the interactions between different entities within a social network take place (usually on site or in the office) and observe and map who goes to whom in search of help or advice regarding work related matters.

In step 2 (data preparation), collected data is cleaned and placed into appropriated formats so that they can be further processed. Usually, data is placed in a matrixial form.

In step 3 (data processing), cleaned and organized data are analyzed through the application of SNA metrics and basic statistics to quantitatively measure collaborative interactions that were captured in collected and prepared data. The metrics used by the proposed strategic process in this work are illustrated in Table 5. These include the centralization degree index, average in-degree index, reciprocity degree index, and the density [32].

**Table 5.** Graph centrality metrics to apply by the strategic process proposed in this work.

| Dimensions | SNA Metric Description | |
|---|---|---|
| Network Access or Reach | Metric 1: Centralization degree index (undirected) $$C_D = \frac{\sum_{i=1}^{g}[C_{TD}(n^*) - C_{TD}(n_i)]}{[(g-1)(g-2)]} \quad (1)$$ $C_D$ = Centralization degree of a network<br>$C_{TD}$ = Total degree of a given entity in a network<br>$g$ = Number of entities in a network<br>$n^*$ = Entity with the highest in-degree in a network<br>$i$ = Entity 1, 2, 3, . . . , $g$<br>Metric 2: Average in-degree index (*directed*) $$ID_{AV} = \frac{\left[\frac{\sum_{i=1}^{g} C_{ID}(n_i)}{g}\right]}{(g-1)} \quad (2)$$ $ID_{AV}$ = Average in-degree index<br>$C_{ID}$ = In-degree of a given entity in a network<br>Metric 3: Reciprocity degree index (directed) $$R = \frac{2L^B}{\sum_{i=1}^{g} C_{ID}(n_i)} \quad (3)$$ $R$= Reciprocity<br>$L^B$ = Total number of reciprocal links within a network | Objective: Analyze the reach (also known as access) of the whole network. It quantifies how much or little access does individuals have to other individuals within an organizational social network function of their need or desire to improve the way they carry out their work. For this matter will be calculated the centralization degree (1), average in-degree (2) and reciprocity (3) indexes. The centralization index varies from 0 (minimum) to 1 (maximum), where 0 means that a network is totally not centralized around one entity, and 1 means that the network is fully centralized around only one entity. The average in-degree characterizes how many links do entities of a network have in average toward the other entities of that same network. It varies from 0 (meaning that there are no links between any entities of a network) to 1 (meaning that all entities of a network have a link to the remaining entities of that same network). The reciprocity index characterizes how many bi-directional links exist within a social network. It varies from 0 (meaning that there are no reciprocal links) to 1 (meaning that all links are reciprocal). | |
| Trust | Metric 1: see (1)<br>Metric 2: Density (*undirected*) $$D = \frac{2L}{g(g-1)} \quad (4)$$ $D$= Density<br>$L$ = Total number of all links within a network | Objective: Analyze the trust level of an organization social network, and thus deduce its knowledge-transfer potential, once interactions such as providing help, support, advice, information transfer take place. This happens more efficiently if the element trust is present. | |
| Communication | Metric 1: see (1)<br>Metric 2: (4) | Objective: Analyze the frequency of and quantify the number of communication channels of an organizational social network. For this matter will be calculated the density of an organizational communication social network. The density varies between (1) and (0), representing a full connected social network regarding the communication channels and the inexistence of any communication channel, respectively. | |

In step 4 (analysis of results and decision making), once having quantitative results from the previous step, decisions are to be taken to either intervene (change) in the actual evolution of the collaborative patterns or to support them. In this step, individual meetings are conducted as a follow-up investigation to better seize the intervention.

In step 5 (apply changes and follow up), decisions taken in the previous set will be implemented, recorded, and communicated.

In step 6 (monitoring and feedback), the implemented actions in the previous step will be monitored and the results of such measures (efficacy and efficiency) will be used as feedback to improve the data collection methods and the following steps. The whole process above described (six steps) is to be repeated in every assessment done in the social manufacturing network.

## 4. Case Study

### 4.1. Introduction

A food and beverage market leader organization applied the proposed strategic process proposed in this work to identify potential collaborative risks in one of its many SCNs between 2019 and 2020 that covers South America and Europe run by two outsourced organizations denominated O1 and O2 due to protection and legal issues. Due to protection and legal aspects, the food and beverage market leader organization will be denominated as organization A (OA) from now on in this work. The SCN of OA where the strategic process was applied is due to the same reasons, from now on in this work denominated SCN 1-2 (SCN1-2) because it extends to two organizations (O1 and O2). The period of application of the proposed strategic process covers a time period before and during the COVID-19 pandemic outbreak. The reason for the application of the proposed strategic process by OA in SCN1-2 is related with several issues that have been emerging within SCN1-2 from 2017 to 2019. These issues have been affecting the whole production chain of a special heat exchanger module for beverage and food purposes. More concretely, such issues were related with several increasing delivery delays caused by several production defects. OA conducted a study through the assessment of a Pulse Survey to analyze the origin of such issues and identified and classified them as problems related to coordination and collaboration within SCN1-2. The expectations of OA as it applied to the proposed strategic process proposed in this work was to improve the resilience and sustainability of the SCN1-2 in hopes that this would reduce extra-production costs due to the high number of defects and improve the collaboration level within SCN1-2. The SCN1-2 of organization OA is illustrated in Figure 5.

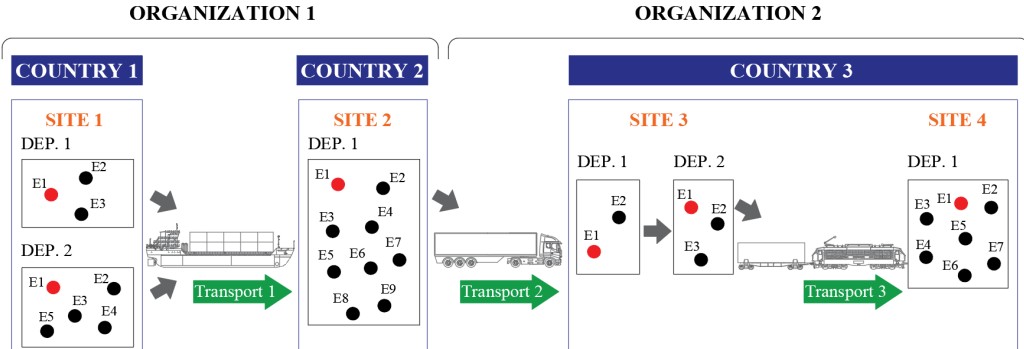

**Figure 5.** OA's SCN1-2.

As it can be seen in Figure 5, the SCN1-2 of OA used to produce the special beverage heat exchanger has four different production sites divided into three different countries, all under the control of OA (multiple-organization multiple-site arrangement according to Table 1), however run by two outsourced organizations (O1 and O2). O1 runs the transportation from country 1 to country 2 as well as the sites of those countries. O2 runs the transportation from country 2 to country 3 as well as the sites of that country.

In country 1, site 1, department 1 are produced the special plates of the heat exchanger module wherethrough will be pasteurized the different beverage end-products of OA. In department 2 are produced the frames where the special plates will be assembled. As illustrated in Figure 5, department 1 of site 1, country 1, has three employees (E1, E2, and E3) responsible to produce the special plates. E1 is the department manager. Department 2 has five employees, and E1 is the manager. Once the plates and the frames are ready, they will be shipped to country 2. In country 2, there is only one site and one department where the parts will be assembled and tested. This department has nine employees. Once the heat exchanged in assembled, it will be sent by truck to country 3. In country 3, there are two sites (site 3 and 4). In site 1, department 1 (two employees), the heat exchanger is received, tested a second time, and chemically treated. Once ready, it is sent to department 2 (three

employees) to be labeled. The last stage before the heat exchanger ready to be delivered to the final customer is site 4, in department 1 (seven employees), where a pre-commercial test will be performed with the respective final beverage solution. Once all stages are successfully concluded, the equipment is ready to be delivered to the final customer.

In order to identify potential collaborative risks across SCN1-2, OA collected collaborative data through the application of a survey to all employees of the SCN1-2. The objective is to map three critical collaborative dimensions as illustrated in Table 3.

To map the first dimension ((1) network access or reach), the following question was addressed: whom would you like to have more access to (be able to talk, brainstorm, and so on), to discuss general or specific work-related matters, under the assumption that if you had access/or more access to that person or persons(s), your performance and the performance of the production chain of heat exchangers would possibly increase, and potentially improve the whole SCN1-2?

To map the second dimension ((2) network trust), the following question was addressed: whom do you feel safe and comfortable to discuss advanced work-related matters, such as suggesting general and/or particular improvements to the SCN1-2 or exchange sensitive information, having the assumption that what is discussed will very likely be taken into consideration (potentially applied and/or scaled to a higher level in the organization´s hierarchy) without fearing any type of rejection or retaliation?

To map the third dimension ((3) network communication), the following question was addressed: whom would you communicate with in a considerable frequency (at least two to three times a week) to discuss general and/or specific work-related matters of SCN1-2?

The assessment was done in three different points in time. The first point in time ($t0$), corresponds to the first assessment ever done in SCN1-2 and is denominated as *period 0*. The exact quantification of *period 0* is impossible to be calculated because it represents a period that theoretically initiates at the very start of the of the SCN1-2. After assessment 1 in $t0$ has been conducted, OA, following the proposed strategic process steps as illustrated in Figure 4, implemented a set of measures to manage potential collaborative risks. The second point in time ($t1$) takes place 14 weeks after the first assessment (denominated as *period 1*). Here OA conducted a second assessment like the first assessment to the social network of SCN1-2 and implemented again a set of measures to manage identified potential collaborative risks. Finally, the third point in time took place 18 weeks after the second assessment. Here OA conducted the last assessment to the SCN1-2 to monitor the evolution of implemented measures to manage identified potential collaborative risks. In the next section will be illustrated and discussed the results of the three assessments.

### 4.2. Application of the Strategic Process

Once all required data for the functioning of the proposed strategic process has been collected, a careful and detailed interpretation of the results took place by analyzing the resulting collaborative networks of each one of the three mapped dimensions.

In Figure 6 is illustrated the results of the three different points in time assessments (CP1, CP2, and CP3) regarding dimension 1-network access or reach.

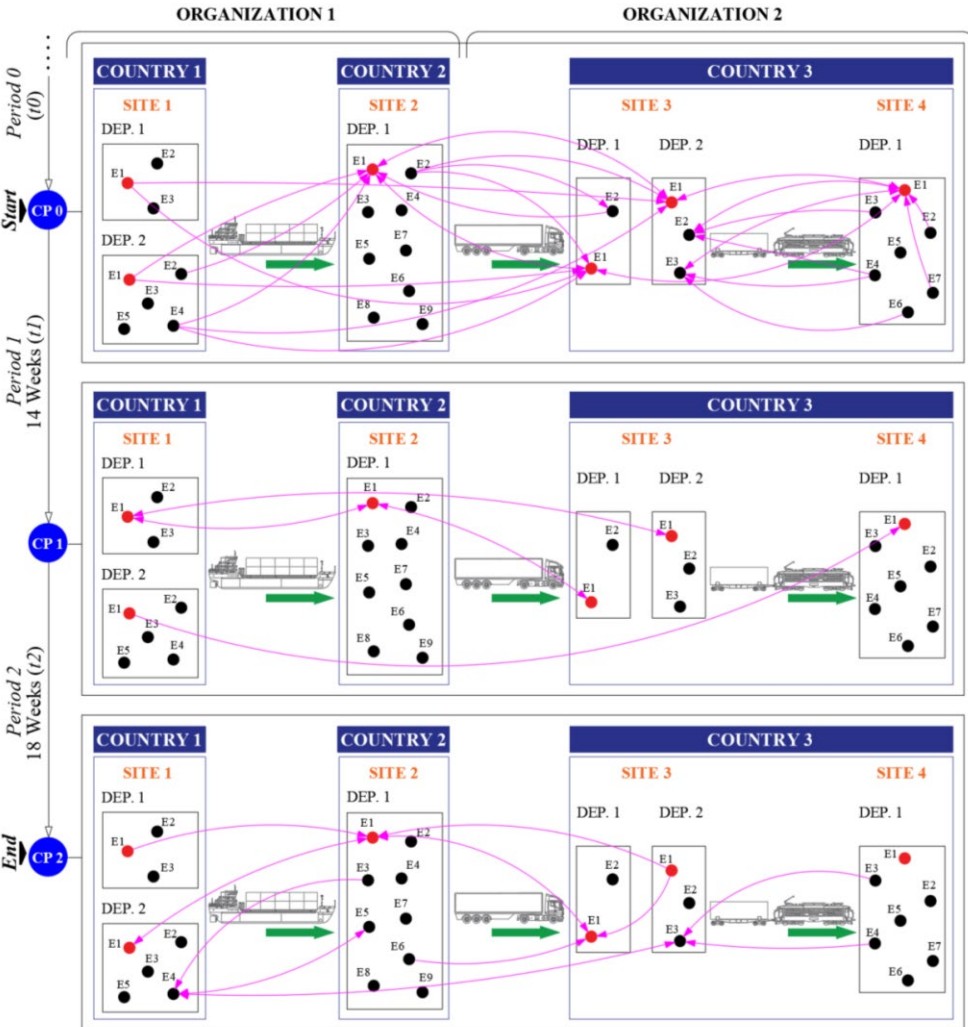

**Figure 6.** Case Study-*More Access To* network.

As can be seen in Figure 6, the resulting networks regarding the *more access to* dimension are different across time. In the first assessment (*CP0*), there are clearly more links between the elements that work in the SCN1-2 than *n CP1* and *CP2*. The resulting network of the first assessment represents, to a certain extent, the working culture of SCN1-2. In *CP0* can be seen that essentially, managers from countries 2 and 1 were nominated as the ones that the SCN1-2 would like to have more access to. Moreover, there is almost no desire to access elements within each one of the departments, except in country 3 where E1 of site 4 was nominated by two internal elements of department 1. Some link towards some nonmanager elements, essentially in site 3, where elements E2 (from departments 1 and 2) and E3 (from department 1) are nominated in the more access to network.

The department 2 of site 3 is by far the one where most links go in, especially coming from site 4, which clearly represents that there are some difficulties to access elements E2 and E3 of department 2 in site 3 from some elements of site 4. This represents the potential existence of collaborative risks within the SCN1-2 that OA should investigate to understand why there is so much need (dependence) of access to elements E2 and E3 from site 4. Still, such mapped state, in the long run, may lead to the emergence of lack of information exchange and consequentially lead to information bottlenecks, which in turn may lead to production delays but also the emerging of production defects and low quality. Ultimately, such state, may lead to the emergence of individual work overload that in the worst case may result in burnouts and other psychological and/or individual physical damages, which will end by affect the whole SCN1-2 [1,33].

While research shows that some amounts of stress (and with very low frequency short-lived acute stress) are good to push people to an optimal level of alertness, cognitive and behavioral performance, in the long term it may be extremely dangerous for the individual, simultaneously putting at risk the team performance [46].

Furthermore, the CP0 network shows the presence of organizational silos. For example, in site 1, none of its elements was nominated in the more access to network. This may represent that either all the other elements have easy access to all the elements of site 1 (which would be the theoretical optimal arrangement) or that there is a lack of need of interaction from the other elements in SCN1-2 with the elements of site 1 due to the "ignorance" of their existence or the apparent low importance that the other elements put on them within the overall production process of SCN1-2. If the case is due to "ignorance" or apparent low importance, it may lead to the emergence of an organizational silo (site 1), which would result in very low collaboration that would special difficult change initiatives of SCN1-2, if needed.

On the other side, there seems to be no difficulty or desire to access any of the elements of site 1 within the social SCN1-2.

Facing this scenario, and after a set of follow up individual interviews, OA identified some issues that needed to be addressed and decided to implement some measures to facilitate the access to some elements of the SCN1-2 social network, namely those with high nomination degree, hoping with this that the SCN1-2 social network become harmonized and less centralized regarding some nominated elements with respect to the more access to dimension, and thus improve collaboration (information sharing, problem solving and so on) of SCN1-2.

After 14 weeks of the first assessment (CP0), OA conducted another assessment to measure the greater access to dimension in SCN1-2. This second assessment is illustrated in Figure 6, CP1. The results of the second assessment clearly show that there is a big change in the arrangement of the more access to network when compared with the network of the first assessment (CP0). In CP1. there are far fewer links within the SCN1-2, being now exclusively managers nominated by other managers in the more access to network. For example, in department 2 of site 3, the in-degree was reduced from 11 to 1.

This means that there were 11 elements desiring more access to the elements of department 2 of site 3, and now there is only one element (E1 of site 1 department 1) desiring to have more access to E1 of department 2 of site 3. Moreover, for the first time, E1 of site 1 department 1 was nominated in the more access to network. In general, it can be concluded that the implemented measures in the SCN1-2 resulted in big changes regarding the greater access to network tuning down the overall need for access, reducing thus the identified potential collaborative risks in CP0.

Finally, after 18 weeks from the second assessment, OA decided to measure a third time (assessment CP2) the greater access to network evolution in the SCN1-2.

The results of the third assessment are illustrated in Figure 6 CP2. This assessment (CP2) was conducted in mid-2020, in a COVID-19-pandemic context. In this context only, some elements of SCN1-2 worked a few times remotely (essentially department managers). The data collection process for the third assessment was through an online survey, instead of a physical survey as for the CP0 and CP1 assessments. In this last network, it can essentially be seen that the number of links did increase when comparing with the previous network; however, it is still far from the number of links mapped in the first assessment.

Nevertheless, some elements, such as the managers from site 2 and department 1 of site 3 and E3 of department 2 of site 3, turned to be once again the most nominated. The CP2 network results clearly show a tendency towards the initial state (CP0), which to a certain extent represents the re-emerging of the existing working culture in SCN1-2, showing the difficulties of changing a given installed organizational working culture as suggested by several research studies [47,48].

However, it can also be observed that in CP2, the links are more heterogenically distributed. For example, the manager of site 2 not only has links from and to other

elements of SCN1-2 but also from elements E3, E5, and E6. The same can be said from E4 in site 1 department 2. This trend may represent to a certain extent the existence of some psychological safety that fearlessly enables the emergence of connections between SCN1-2 elements which is in line as suggested by some research [1,45]. It may also represent a higher level of interaction between elements of SCN1-2 resulting in more engagement, empowerment, and collaboration.

To better understand the effects of the implemented measures of OA to minimize or/and eliminate collaborative risks in SCN1-2, a longitudinal analysis is illustrated in Figure 7. In Figure 7 are illustrated three metrics (average in-degree index which results of applying (2), the centralization degree index which results of applying (1), and the reciprocity index (3)) results defined in Table 5 for the analysis of dimension 1—network access or reach.

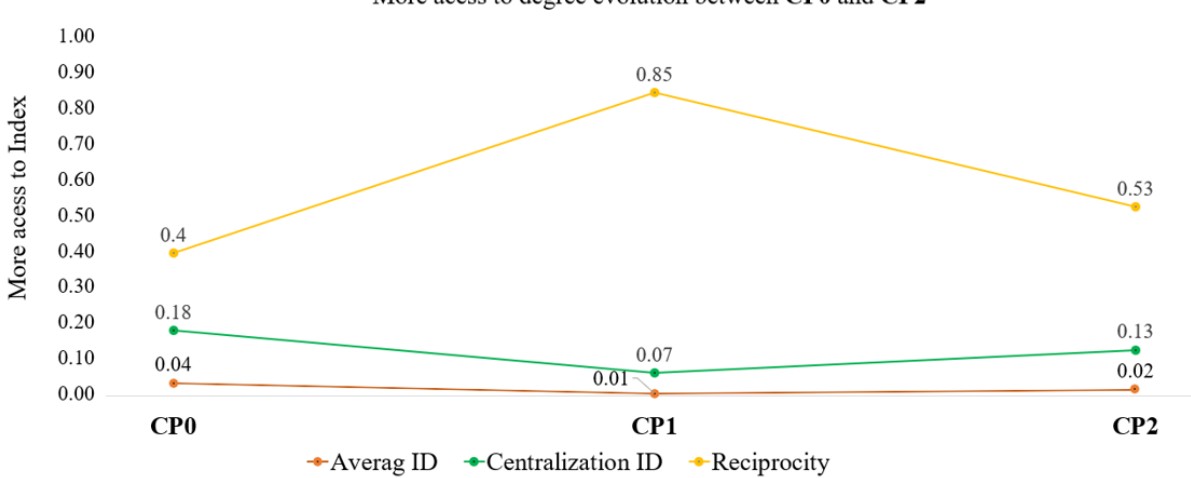

**Figure 7.** Longitudinal analysis Case Study-More access to Network.

As it can be seen in Figure 7 the evolution of the more access to dimension between CP0 and CP1 shows a clear variation.

In CP0, the average indegree has a value of 0.04, which means that is located very near to one of the collaborative extremes illustrated in Figure 2, more concretely *a1* rather than *a2*. This represents, for the case of more access to network, a positive and, to a certain extent healthy, collaborative state, if the value were 0, or 1. However, the value translates that on average every element of SCN1-2 has a value of 1.03 in-links, which is very far from being the truth illustrated in the network of CP0 where 12 out of 30 have no in/out degree (40%) and 21 out of 30 (70%) have no in-degree in the more access to network.

This means that the average in degree is a necessary metric to shed light on what is going on in the whole SCN1-2 regarding the more access to network, but alone it is not sufficient to explain the full picture. Only when applied additional SNA metrics such as centralization and reciprocity, one can grow pretty much closer from the reality. Both centralization and reciprocity are illustrated in Figure 7.

In CP0, the centralization index has a value of 0.18, which is located between the halfway of either heading towards a total individual collaborative overload and lack or inexistence of network collaboration (index degree 1 in Figure 2) or towards the other extreme which is characterized either by a lack or inexistence of network collaboration or a chaos state (index degree 0 in Figure 2), and the latter. This means that the SCN1-2 is not in any of the mentioned extremes regarding collaboration. However, it has a higher value than the average indegree value, which means that although the network has, to a certain extent, a healthy collaborative state, the centralization index says that there is, or there are, some elements emerging with some disproportionality and centrality regarding the greater access to network.

In fact, this is what happens when we analyze the centrality of managers of countries 1 and 2 where the centrality degree varies from five (min) to six (max) in-links, which is four to five times more than the average in-degree of 1.03 in-links. Translating the results illustrated in Figure 7 regarding the centralization index into practical and actionable terms confirms the potential existence of some risks regarding collaboration within SCN1-2 as already mentioned above.

However, the analysis only becomes fully fruitful when the reciprocity index is analyzed. The reciprocity index has a value of 0.4 which represents 40% of all existing connections (30 links) in SCN1-2; more access to network is reciprocal. This means that there are 12 reciprocal connections in 30. This value is very high because it represents that almost 50% of the elements that mean someone has difficulty gaining access also mean that someone is difficult to access. Ideally a reciprocity value of 0 in the more access network would mean that any two given two elements in the SCN1-2 would not need more access to from one each other, rather in one direction only (A need more access to B, but B does not need mode access to A).

On the other side, a value of 1 in the reciprocity index would mean that everybody would need more access from everybody, being that there is a lack or inexistence of network collaboration and simultaneous chaos regarding the need from one to each other. However, a high value of reciprocity is not necessarily a bad thing.

In fact, it can be a great help once we find the unlocking factor(s) that enable(s) us to create the time and connection between any two given elements that require more access from each other, and vice-versa. This new insight given by the reciprocity shows that the collaborative relationship within SCN1-2 clearly has room to be improved. However, it is not a one-way-only road to improvement. Most likely, measures implemented to improve one collaborative dimension may negatively affect other dimesons.

Moving along the X axis, it can be seen in Figure 7 that the average in-degree has decreased from 0.04 to 0.01 in CP1. This means that on average, the SCN1-2 has fewer elements that need more access to other elements. The same trend is observed in the centralization degree from CP0 to CP1. This means that there are also fewer elements with a disproportional centrality within SCN1-2 regarding the more access to network.

However, from CP0 to CP1 the value of the reciprocity increased up to 0.85. Such increase can either be explained by the direct influence of the reduction of the number of links given by the average on-degree. Nevertheless, as previously see, it may be a very good indicator once collaborative risks need to be addressed. In Figure 8 is illustrated the results of the three assessments (CP0, CP1, and CP2) regarding dimension 2 (network trust).

The networks illustrated in Figure 8, unlike the network illustrated in Figure 6, are of the reciprocal type (also known as undirected) because it is assumed (however knowing that reality can differ from this assumption) that if A trusts B, then B trusts A. In CP0, the trust level is by far (as expected) more concentrated within each of the departments rather than between any two departments or any two sites or countries, with some exceptions, such as the case between department 1 and 2 of site 3, and the latter with site 2.

This may be explained by the geographical proximity, and potentially by the principle of homophily, as suggested by serval research [1,29]. Furthermore, the trust network evolves from a siloed state towards a networked state (cross boarder) as we move from CP0 to CP2. This evolution shows the effects of the measures implemented after the two assessments CP0 and CP1, conducted by OA in the SCN1-2.

In Figure 9 are illustrate two metrics (the centralization degree index (1), and the density (3)) results, as defined in Table 5 for the analysis of dimension 2—network trust.

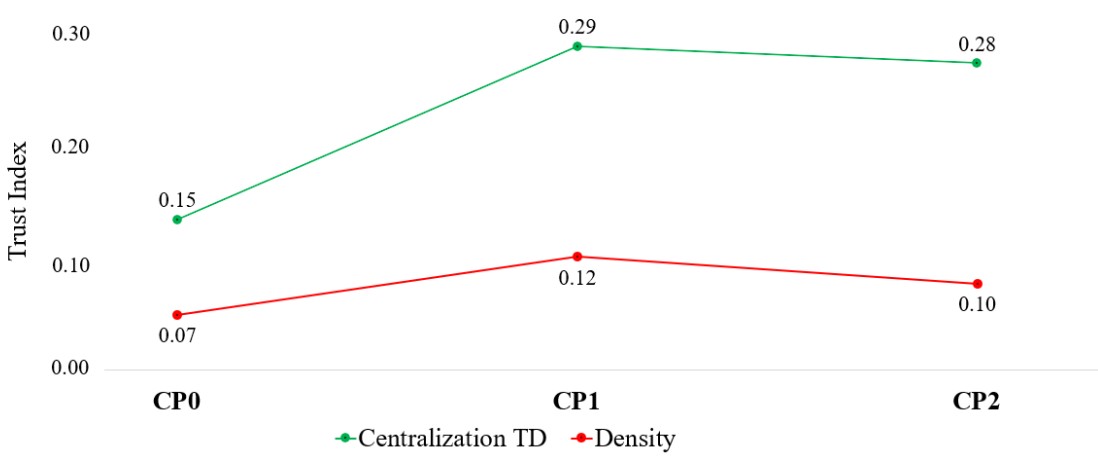

**Figure 8.** Case Study-*Trust* network.

**Figure 9.** Longitudinal analysis Case Study-Trust Network.

Being that trust is the major factor for achieving efficient and high level of collaboration as suggested by several research studies [1,11,38–40,45], the ideal values to be observed in the trust trend line for the centralization degree index and the density would be of 0 and 1, respectively.

This would mean that everybody in the SCN1-2 would trust everybody in a way that they could fearlessly exchange information, discuss ideas, share knowledge and so on. Furthermore, research shows that trust is the critical element that must be strong within a given social network so that a psychological safety environment around every participant can emerge, and thus potentially originate high performance teams [1,45].

Trust within a social network is a critical element that increases information exchange in a valid, transparent, and efficient way, enables the creation of efficient problem-solving networks, and generates well-being and belonging [8,9].

However, such desired values are not present in Figure 9. In fact, the value of the trust index measure through the centralization metric at the beginning in CP0 is somewhat high (0.15). This value shows the existing level of trust until the first assessment, which is the same as saying that such level of trust reflects a given working culture state that exists for very long time in SCN1-2 and means that there is some centralization in the trust network which translated the existence of some elements that have many trust links when compared with the rest of the elements. This is the case of for example elements E1 in site 2 and in department 2 of site 1.

Such level of trust puts at risk all the benefits above-mentioned of having a high level of trust within a given social network, and thus may be one of the reasons that is leading to the emergence of collaborative problems within SCN1-2, which can be translated into poor overall performance.

However, after the implementation of some measures by OA, the level of trust clearly improved, according to the results obtained in the second assessment. The density moves from 0.07 up to 0.12, which represents an increase of almost 50% from CP0 to CP1, but the centralization degree also increases by about 50%. This is explained due to the values of some elements sin SCN1-2 which had a substantial increase in their trust level such as E1 of site 2, E1 of department 2 with 11 and 8 links, respectively.

Such increase is in fact a strong positive signal; however, there may exist some bias in the answers provided by the elements of SCN1-2, essentially influenced by a need of becoming more collaborative as a condition to become more productive and to attain a better performance evaluation.

Such bias can also be observed in the other mapped dimensions, but not in the same intensity as in the trust network due to the nature of que question addressed to the participants.

In fact, as we move along the x axis from CP1 to CP2, all the trust indicators decrease. However, while from CP1 to CP2 the trust level decreases regarding cross-boarders (between different departments, sites, or countries) the trust level within the departments tends to stabilize in the values of CP1. In Figure 10 is illustrated the results of the three assessments regarding dimension 3 (network communication).

As it can be seen in the network of CP0 illustrated in Figure 10, the communication in a high a considerable frequency between the elements of SCN1-2 is to a certain extent weak, but it improves as we move in the x axis, as can be seen by the increase of links between the elements of SCN1-2. In CP0, there is no communication with a considerable frequency between site 1 and site 2 and site 4, a trend which is also to be observed in the trust network. Such observed state in CP0 may largely contribute to the emerging of collaborative issues such as organizational silos and lack or total inexistence of collaboration. In fact, as research shows, if there is no interaction whatsoever, there will never exist room for creation of trust [9,38,39]. Research says that communication is the veins of a social network, wherethrough vital information flows, and that performance is simply a function of how efficient such veins are [1,9].

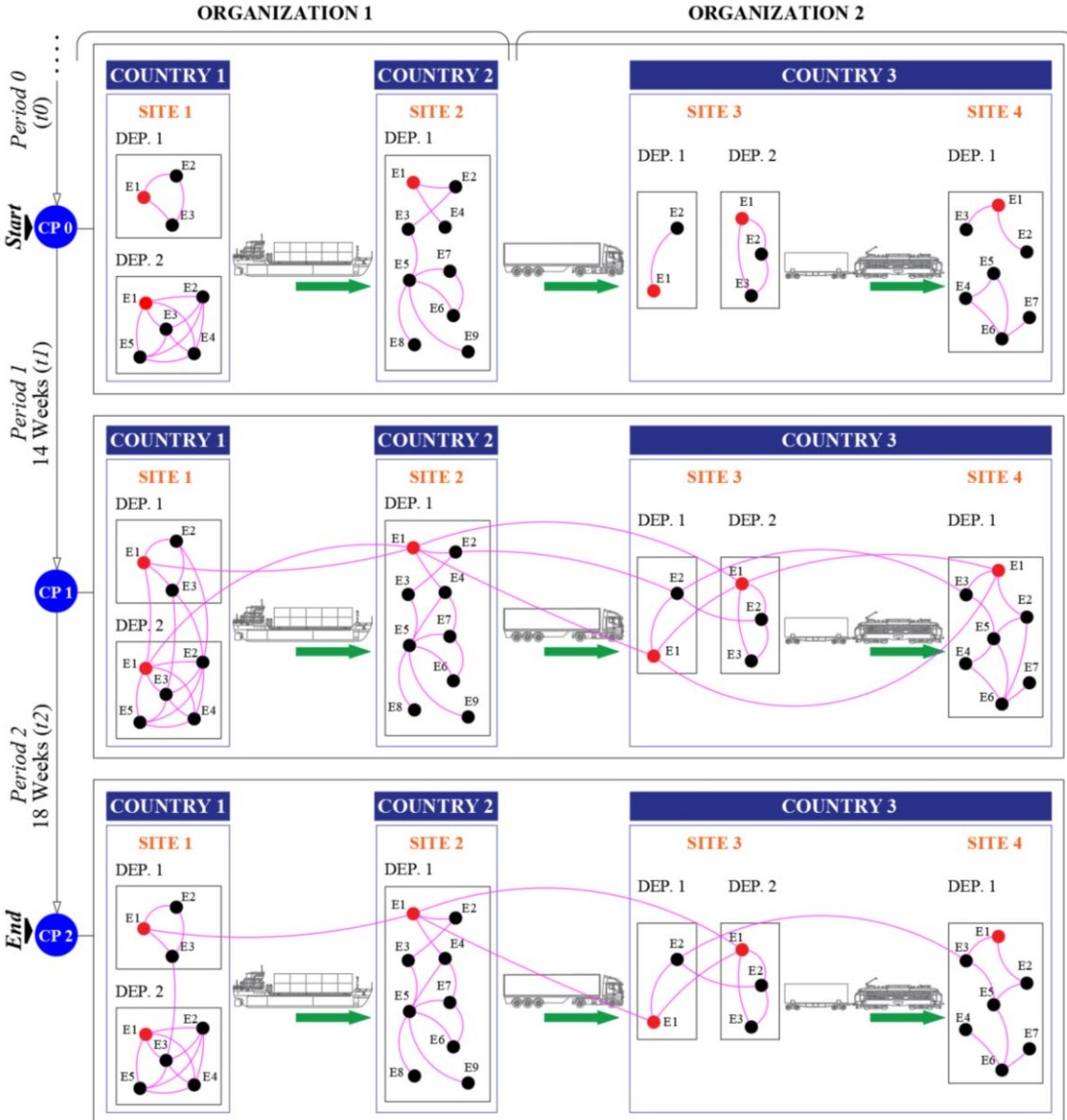

**Figure 10.** Case Study-*Communication* network.

One interesting aspect to be observed in Figure 10 is that there seems to be a shift regarding the central elements within the network as we move along the x axis between CP0 and CP1 when comparing with the other two previous networks. For example, in CP0, E5 of site 4 has a higher degree of communication than E1. In CP1, E3 of department 1 in site 3 has as many links as its manager.

However, as we move towards CP2, there is a certain decrease in the number of links of nonmanager elements when compared with their managers. Such trend, similar with the trend of the trust dimension, is observed as the number of links cross-orders decrease, but the links within each one of the departments remains unchanged. In Figure 11 are illustrated two metrics (the centralization degree index (1), and the density (3)) results, as defined in Table 5 for the analysis of dimension—communication.

Communication degree evolution between **CP0** and **CP2**

**Figure 11.** Longitudinal analysis Case Study-Communication Network.

As it can be seen in the trend lines in Figure 11, although the communication degree increases across the x-axis in CP1 and CP2 when compared with CP0 (to be observed in the density values), the evolution characterized by the centralization and the density does not vary significantly across time. In fact, in a general view, it can clearly be seen that the measures applied by OA after CP0 and CP1 contributed to simultaneously lowering the centralization degree and increase the communication channels between the elements of SCN1-2.

Such evolution arrangement is in fact very positive; communication should not be disproportionally increased because one risks landing into a state of communication chaos, which may be considered as bad as having no communication at all. To better understand the evolution of the three dimensions across time and how the measures that O1 applied to balance and bring back the SCN1-2 to a healthy collaborative state impacted the collaborative patterns of elements of SCN1-2, a longitudinal analysis is illustrated in Figure 12.

Evolution of the three dimensions between **CP0** and **CP2**

**Figure 12.** Evolution of the three dimensions between CP0 and CP2 regarding the centralization degree index.

As can be seen in Figure 12, the evolution of the three dimensions ((1) more access, (2) trust, and (3) communication) have very distinct evolutions across time within SCN1-2.

Such evolution was impacted by the measures applied by OA to correct, align, or support some identified behavioral patterns. In Figure 12 is clearly visible the impact of the measures applied to lower the centralization degree in the greater access to network. For example, the dotted line in Figure 12 that represents the centralization index evolution of the greater access to network moves from an initial 0.18 to 0.07 and finally lands in 0.13. This evolution is positive because it decentralizes the greater access to network lowering the need for access level from some elements towards other elements of SCN1-2. Simultaneously, the measures implemented by OA enabled to increase the trust level (short-dashed line in Figure 12) within the SCN1-2 rising from a trust index of 0.15 to 0.29 and landing in 0.28. This achievement is by far the most important improvement that the propose strategic process enabled to O1 to achieve. It represents an increase of 62% of trust within SCN1-2. Being that trust such a fundamental element of a social network, this achievement can only positively impact the collaborative capacities of SCN1-2. Finally, the communication dimension (long-dashed line in Figure 12) has also suffered some improvements; however, it is the dimension where apparently the measures applied by O1 had the smallest impact evolving from a 0.11 index to a 0.10 and landing in a 0.12.

After the assessment, individual follow up interviews to identify root causes of identified behavioral patterns have been conducted and resulted in several measures to implement to improve resilience and sustainability. For example, the introduction of a 45-min *cross-border virtual talk-time* program, running twice a week with alternating schedule (due to time difference between the different three countries) aims to create new communication channels—namely cross-boarder channels that connect any two sites, departments, or countries—by introducing the discussion of new ideas, challenges and problems that are emerging across the whole SCN1-2.

Another initiative under the name *out-visit* takes place every three months and consists of putting some elements from a department or site of the MN in a different department or site of SCN1-2 so that they can observe how work is done in a different site or department than theirs. By doing so, the SCN1-2 social network is gaining awareness of who does what and where and enables elements of SCN1-2 to efficiently identify whom they really need to talk more to, contributing thus to lower the centralization degree of the more access to network. The reallocation of some tasks within the SCN1-2 is another initiative implemented by O1, and it consisted of allocating some elements in the right workplace, at the right time, doing the "right work" (finding who is best "sized", if possible, to execute a given work task or activity), so that work can be done better. This also enabled us to lower the centralization degree of the greater access to network, as elements of SCN1-2 were waiting to gain access to certain elements to receive help to finish their job. The almost complete *digitalization* program conducted by OA consisted of a huge increased use of remote work and communication tools, especially when the COVID-19 pandemic broke out. This enabled some work to be executed in a faster and more integrated way, and it facilitated the communication within SCN1-2, namely the cross-boarders communication.

The introduction of the *task redundancy* (which consists of having a set of elements able to execute a given task or activity) and *profile redundancy* (which consists of having a set of elements with the same accesses to programs, procedures and contacts) programs, enabled SCN1-2 to create a strong and actionable know-how backup (resilience) in order to avoid know-how and now-what leakages in case one key element or elements suddenly for any reason must be out of office or leave the organization for a considerable period of time. Such key elements were identified in the greater access to network (essentially those with a considerable in-degree).

In the sustainability side, OA introduced the *be-sustainability* program, which was possible to implement, essentially due the improvements done by the implementation of the programs above-mentioned. Instead of hiring new people into the SCN1-2, because the assessment conducted, it was possible to improve the overall performance of SCN1-2. The *be-sustainability* program consists of reducing the working time in 1 h a day for a one-week period (maintaining all financial benefits), continuously alternating across all elements of

SCN1-2. By doing so, OA estimates that there is a reduction around 900 kg of $CO_2$ footprint per year because 1 h at office consumes about 450 Wh, that one labor-year has 261 days, and to produce 1 Kwh costs about 0.94 kg of $CO_2$ emissions. Such reduction (900 kg of $CO_2$) is equivalent to drive about 7000 Km per year with a combustion engine car.

Finally, the *be-sustainability* program consisted also of creating a sustainability culture across the SCN1-2 and beyond (to the extent possible). It consisted of persuading SCN1-2 elements to be more aware of climatic challenges and how they could contribute to tackle them in their daily-work routines. To influence the whole SCN1-2 in the most efficient way possible, OA named some SCN1-2 key elements as sustainability ambassadors. Such key elements were identified by merging the trust and communication networks and identifying the elements with the highest total and brokerage (access cross-boarder's elements) degrees. Such elements are those that have a higher capacity of influencing others within SCN1-2. By doing so, OA strongly increased the message dissemination speed as well as its importance (people will take it more seriously because they have been informed by someone they trust).

## 5. Conclusions

In this work is presented a strategic process to manage collaborative risks in SCNs to improve resilience and sustainability.

The proposed strategic process is developed based on three pillars ((1) supply chain networks, (2) risk management, and (3) social network analysis) and analysis of the evolution of three key SCNs collaborative dimensions ((1) network access or reach, (2) trust, and (3) communication) looking for behavioral patterns that potentially may cause some of the issues that lead to poor performance and thus threaten the resilience of the collaborative dimension, which in turn may threaten the chances of achieving goals and objectives and the survival of one organization.

The propose strategic process aims to act essentially in the first stage of the four stages of the resilience mechanism—the avoidance stage as illustrated in Figure 13. For the illustration of Figure 13, only the four major resilience stages have been adopted [21].

In Figure 13 are illustrated the four major stages of a typical organizational resilience mechanism (avoidance, containment, stabilization, and return) as well as some of the strategies that organizations must manage in each one of the four stages. As mentioned, the major objective of the propose strategic process proposed in this work is to act in the avoidance stage, which in the context of collaborative risks means exactly to avoid heading to one of the two collaborative extremes (Figure 13a) denominated as lack of collaboration and/or chaos (left side in Figure 13a), and collaborative overload/high dependency (right side in Figure 13a) that may emerge as the result of any disruptive event (when the disruption occurs in Figure 13a—from Figure 2). By applying the strategic process proposed in this work (SP in Figure 13b), an organization very likely avoids the heading toward one of the collaborative extremes. However, if that takes place, the strategic process can also be beneficial to organizations to understand the root causes that led to such disruption in the collaborative dimension.

Some of the strategies (and the respective effects) that the strategic process is comprised of to efficiently deal with each one of the stages an organization is regarding in a collaborative dimension are illustrated in Figure 13c. These strategies have been concluded through the application of the strategic process model in several organizational scenarios across several years. For example, in the avoidance stage, the strategic process recommends the use of task execution redundancy to avoid the emergence of task-execution bottlenecks for example. This strategy has a high impact to avoid the emergence of such task-execution bottlenecks. In the avoidance is the creation of a flexibility mindset across a given SCN social network that has a low contribution to avoid entering one of the collaborative extremes.

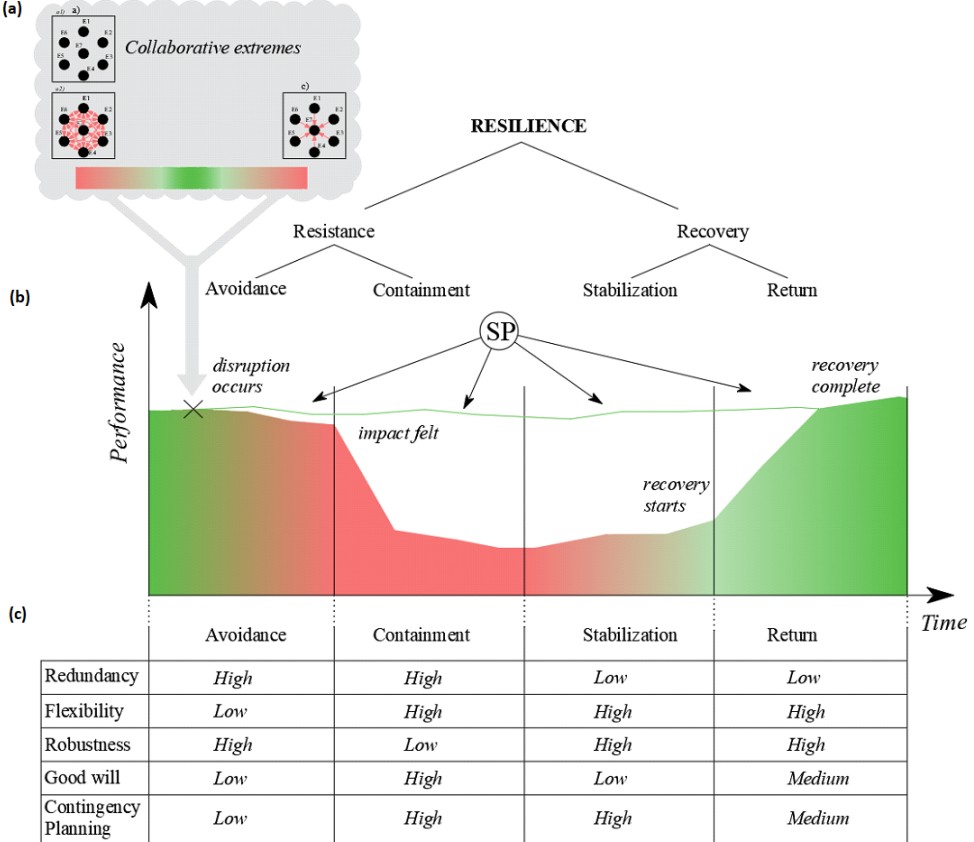

**Figure 13.** Typical resilience stages and resulting strategies to enhance collaborative resilience, adapted from [21]—(**a**) two collaborative extremes according to Figure 2, (**b**) relationship between performance and the different stages of organizational resilience, (**c**) measures to address the different stages of organizational resilience and their effectiveness.

In the illustrated case study are clearly demonstrated the contributions of the proposed strategic process to enhance the resilience of the collaborative dimension of organizations that work in SCNs. This can be seen across the case study where the application of the proposed strategic process enabled OA to identify and manage collaborative risks in the SCN1-2 used to produce special heat exchangers modules. Such collaborative risks were threatening the successful production process of SCN1-2, as the performance results were being released across time. Before the application of the proposed strategic process SCN1-2 was having several issues in the production chain, which were essentially translated into several delays and production defects in one side, on the other side was emerging a nonhealthy work environment among the elements of SCN1-2, which was leading to several complains to the management team. Such facts, in such competitive and instable market landscape, could particularly damage, or even kill SCN1-2 of OA.

The application of the proposed strategic process successfully helped OA to first understand to which extent collaborative patterns among the elements of SCN1-2 that have been evolving across time, could be negatively influencing the whole production chain. Second, the proposed strategic process successfully enabled OA to properly develop measures to bring collaboration back to track, and thus potentially recover from what has been lost.

On the upside of the implementation and application of the proposed strategic process in this work, some points can be highlighted. First, all the process around the strategic process was well received and accepted by all the elements of SCN1-2. Second, the production efficiency (measured in delays and in the number of heat exchangers produced with defects) increased to 30% the after the implementation of corrective and adjusting measures.

Third, the introduction of redundancy was well received and enabled more interaction and information sharing among the elements of SCN1-2. Moreover, there has been a reduction in the number of email communication namely in providing feedback since the SCN1-2 has started to use with much higher frequency virtual on-time communication tools that enabled us to treat feedback activities in a much faster way. Finally, the SCN1-2 social network gained notable increase in the social well-being aspects, which lead to strongly reducing the number of complaints received by SCN1-2 managers.

On the downside, some of older elements of SCN1-2 felt that they were being over-monitored by the application of the proposed strategic process which consisted only of answering three questions in two points in time and the respective follow up interviews and implementation of measures, and that by doing so, OA was not entirely trusting his outsourced organization´s 1 and 2 employees. Some elements of SCN1-2 also felt a bit uncomfortable with the reallocation of some work tasks and activities. This essentially happened because the task reallocation lowered the workload of some elements of SCN1-2 to half, and others increased their workload. Finally, although the second assessment (CP2) was done in a Covid-19 context, the outbreak to the COVID-19 pandemics did not affect much the SCN1-2 performance and resilience; rather it affected their external SCN. This was possible because right before the outbreak, OA had implemented the already mentioned corrective measure that helped to face the unknown upcoming times in a much more prepared way.

## 6. Academic, Managerial and Ethical and Legal Implications

### 6.1. Academic Implications

The proposed strategic process in this work addresses a very interesting, intriguing, and rich subject that exists in the great majority of all organizations—collaborative risks. The proposed strategic process identifies collaborative behavioral pattern in SCNs bay analyzing three critical collaborative dimensions ((1) network access or reach, (2) trust, and (3) communication) that exists in most organizations. The development and application of the proposed strategic process in organizations contribute to the development of each one of its fundamental pillars ((1) supply chain networks, (2) risk management, and (3) social network analysis). Regarding the first pillar, the proposed strategic process contributes to better understanding the importance of the efficient management of potential collaborative risks in SCNs, but also in SCNs. This is supported with several research studies that indicated the coordination and collaboration management in SCNs as one of the principal areas of ongoing research [12–14,21].

Regarding the second pillar, the proposed strategic process contributes to development of standard-based risk management models adjusted to specific environments as is the case of the adoption of the ISO 31000 standard (ISO 31000) to strategically process the whole approach of the proposed model. Regarding the third pillar, the application of the proposed strategic process enables us to support or contest in a more data-informed way research that highlights the fundamental role of organizational behavioral patterns in innovation and performance [1,40], and research that argues that other factors, such as education, business referral and expertise, are of greater importance to boost innovation and performance, and thus increase resilience and sustainability [49,50].

By focusing the analysis on the dynamic interactions between elements of a social network (also known as behavioral patterns), the proposed strategic process provides unique contributions to the corporate behavioral scientific field, which according to research is still very underdeveloped [19,40].

Proposed strategic process in this work is aligned with latest research [51] that shows that collaboration works better under a hands-on approach (more control of the dynamic interactions of organizational social network´s elements) rather than a hands-off approach (leave the management of organizational collaboration at chance) because it represents a tool that helps to exert more control in a constructive way over the interactions of organizational social network´s elements.

Finally, the proposed strategic process positively contributes to the organizational transformation process (digitalization) in the sense that it provides organizations a new approach to manage risk (collaborative risks) across the different departments, sites and geographic locations of a manufacturing network and the different phases of a project lifecycle by the application of information and technology tools and approaches as it is the incorporation of the proposed strategic process in a business intelligence architecture. Such transformation requires the implementation of new technologies across an organization's structure and the adoption of a new ways of working, as proposed by several research studies [7,19,25], which ultimately could lead to the development of new organizational theories and approaches on how to manage organizational collaborative risks in a more predictive way.

*6.2. Managerial Implications*

Throughout this work has been demonstrated how the proposed strategic process can help organizations to efficiently identify and manage their organizational collaborative risks. The proposed strategic process uncovers collaborative blind spots in the collaborative dimension in a very effective way, which in turn enables organizations to improve their flexibility, change, and improve their risk assessment processes. This is aligned with much research that shows collaborative dimension's problems and issues are the top priorities in manufacturing and SCNs organizations [4]. The application of the strategic process to manage collaborative risks enables organizations to better protect their worker´s physical and mental safety, as for example in the timely identification of the emergence of disproportionally central elements of a manufacturing network, which in turn may become bottlenecks and possibly result in personal and organizational damages. This way, the strategic process helps organizations to ensure business continuity and simultaneously contributes to improve individual performance because it clearly identifies two central elements which, for example, due to their multitasking capacity, accumulate several tasks or activities decreasing execution performance as suggested by several research [1,52]. The proposed strategic process also has been proven efficient even in unexpected times such as those of the COVID-19 outbreak. In this line of thought, the strategic process helps organizations not only to prepare the short-term but also the long-term response of an organization to unforeseen and unexpected disruptions.

The proposed strategic process provides organizations an integrative tool that enhances their overall risk management processes. For example, measuring workload in terms of network relationships between employees of an organization is not so accurate as the well-known Pulse Surveys where people are asked to answer a set of questions regarding how they experience work tasks and activities, and later quantitatively measured using a Likert scale, for example. However, an assessment to a given organizational social network using the proposed strategic process after knowing the results of a Pulse Survey—where individual opinions regarding work matter related are characterized—will shed light into most of the real root causes that are behind the individual states [29].

The application of the strategic process enables organizations to create efficient redundancy of execution of work tasks and activities so that the business ban continues when key elements must leave for any reason (as demonstrated in the case study).

Because the proposed strategic process quantitatively identifies collaborative behavioral patterns, organizations can better understand and efficiently correlate different collaborative trends with different outcomes. This, in turn, enables decision-making processes to be more supported on factual data, rather than exclusively relying on gut feelings and key influencers' opinions.

The proposed strategic process provides organizations a unique and valuable tool to, in a quantitative way, identify hidden collaborative behavioral patterns, which according to research [1,29,38–40] cannot be understood and managed by the application of traditional project management tools and techniques.

The successful application of the strategic process also contributes to the achievement of some objective defined in the sustainable development goals published by the United Nations [53] such as the good health and well-being in the workplace, industry innovation and infrastructure, sustainable cities and communities and finally responsible consumption and production.

Finally, the application of the strategic process contributes to improve the social network mental health and psychological safety, being considered one of the most important factors before, during and after the COVID-19 pandemic [18,29], preventing the emergence of two collaborative extremes that strongly hinder resilience and the achievement of sustainability objectives, leading to total lack or inexistence of collaboration within a given manufacturing network, or total collaborative overload as illustrated in Figure 2.

### 6.3. Ethical and Legal Implications

The proposed strategic process in this work accesses and analyzes what by many organizations can be classified as sensitive and confidential work-related information that flows across the different manufacturing network´s elements, which in some cases may not be accessed and/or exposed. Therefore, the implementation and application of the proposed strategic process in this work is fully dependent on the acceptance of the competent authorities at both organizational and national levels that administer the legal and ethical respective issues, as is the case of the GDPR (General Data Protection Regulation) regulations, applied in European countries [54].

Furthermore, all the elements of a given manufacturing network should be aware in advance that work behavioral information will be accessed and analyzed for controlling and monitoring collaborative patterns so that the organization can improve performance and thus become more resilient and sustainable.

### 7. Suggestions for Future Research

The implementation and application of the proposed strategic process may represent a challenge for some organizations because they may not yet have the necessary technologies and/or working culture that enables the proposed strategic process to efficiently identify collaborative risks.

In this line of thought, it is suggested that organizations must first create an organizational architecture (for example the integration of the proposed strategic process in this work into an organizational business intelligence architecture), where data can be collected, stored, and later analyzed. Becoming a data-literate organization—which is characterized by the ability to understand, engage, analyze, and reason with data—is still a challenge for most organizations. For organizations to create value, capabilities, and make smart and timely business decisions, data must be first democratized (accessible to everyone, and bottlenecking-free, except if it is considered confidential or highly sensitive), normalized (standardized, i.e., same values, expressions, language and so on), reusable across different applications and geographies, and readily available (without time lags).

The integration of the strategic process into a typical business intelligence architecture or ERP system will enable organizations to boost their digitalization initiatives which improves flexibility and agility, but also for improved responsiveness in meeting customer requirements, quality, and continuous improvement as suggested by several research studies [1,55].

The application of other SNA metrics such as betweenness and closeness [32] is also suggested to identify other potential hidden collaborative risks.

Finally, the proposed strategic process may collect data from log files, surveys, meetings, and observations. However, as much work-related information flows across other communication channels, it is suggested that research regarding the data collection process through other channels should be undertaken, such as phone calls, corridor meetings and virtual communication platforms.

**Author Contributions:** Author M.N. carried out the investigation methodology, writing—Original draft preparation, conceptualization, the formal analysis, collected resources, and application. Other remaining authors (A.A., J.B., E.N. and C.S.) contributed with the review, and validation. All authors have read and agreed to the published version of the manuscript.

**Funding:** This research received no external funding.

**Institutional Review Board Statement:** Not applicable.

**Informed Consent Statement:** Not applicable.

**Data Availability Statement:** Not applicable.

**Conflicts of Interest:** The authors declare no conflict of interest.

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
