# Peer review of "A Strategic Process to Manage Collaborative Risks in Supply Chain Networks (SCN) to Improve Resilience and Sustainability"

_sustainability, doi:10.3390/su14095237_

Round 1

Reviewer 1 Report

Your case study is very good. Please support it with good theoretical foundation. See the comments in the PDF.  Perhaps you added the wrong Literature list or something else happended - the numbers and the content do not match for all entries. Since your method is sound and useful, as the case shows, you should revise the article. That should not be too much work but be worth it.

Author Response

Rev 1

Thank you very much for all your comments. We believe that in fact helped greatly to increase the overall quality of the manuscript. All your comments were taken into consideration in the revised version of the manuscript. All changes are marked in red in the new manuscript version.

Your case study is very good. Please support it with good theoretical foundation. See the comments in the PDF.  Perhaps you added the wrong Literature list or something else happended - the numbers and the content do not match for all entries.

Thank you very much for this point. In fact, we found several references out of order. We unfortunately do not know what happened. However, we corrected them across the manuscript as well the formatting in the references section. We also correct the Tables and Figures entries across the document.

Since your method is sound and useful, as the case shows, you should revise the article. That should not be too much work but be worth it.

Thank you very much for this point. This comment is very encouraging to proceed to the revision and improve the overall quality of the document.

There is no consistant story to follow - why should I read about strategic process development right now? What is the connection to collaborative risks? That is not obvious: Explain it!

Thank you very much for this point. We fully agree with this point. In fact, the way the section is named generated some confusion among the reviewers. What we mean there is that the proposed “model “ in the manuscript is simply named as strategic process. So, what we mean with that in section 3 is the introduction to the development and implementation of our proposed strategic process to manage collaborative extremes in SCNs.

Allow the reader to find out, what your conclusions are and what you extracted from literature. Did you draw the sketch? Make that obvious.

Thank you very much for this point. The literature review section has been rewritten and reorganized in order to better explain the reader the background and gaps for the presented research. This topic has been also tackled by other reviewers.

All remaining comments are answered directly in the new version of the manuscript.

Reviewer 2 Report

Abstract. Purpose: briefly describe the aim. Methods: briefly describe the main methods or treatments applied; Results: summarize the article's main findings;  Conclusions: indicate the main conclusions or interpretations.

Show in introduction. What is the purpose of the article’s analysis? The aim of the article needs to be adjusted. Clearly define research questions and answer this questions in the article.  Why is this research important to the reader? What kind of gaps does it cover?

Line 38. Check references style [1,2,3]

Line 56 Check references style [7, 8, 9].

Line 59 Check references style [7, 9, 10, 11]

Line 64 Check references style [4, 12, 13, 14].

Line 66 Check references style [2, 3, 4, 12, 13, 14, 15].

Line 127 Check references style [ 14, 13, 20].

Line 127 Check references Line 315 Check references style [ 1, 33, 32, 42, 43, 44, 40].

Literature review. 

The gap or shortcomings of the literature has not been mentioned anywhere in the literature review section. The authors continue to describe different studies but did not mention the gap in the literature. This section must be rewritten as well. It’s really hard to understand what authors mean by mentioning these studies.

Strategic Process Development and Implementation wrote very long and confusing. It’s really hard to understand what authors want to explain. Add the Materials and Methods section. 

Add source to pictures.

The sections describing  the Case study are described in too much detail. 

Show the data volume requirements. How much data needs to be collected for the estimates to be reliable.

Check references style, depending on the type of work. See the Reference List and Citations Guide for more detailed information.

Author Response

Rev 2

Thank you very much for all your comments. We believe that in fact helped greatly to increase the overall quality of the manuscript. All your comments were taken into consideration in the revised version of the manuscript. All changes are marked in red in the new manuscript version.

Abstract. Purpose: briefly describe the aim. Methods: briefly describe the main methods or treatments applied; Results: summarize the article's main findings;  Conclusions: indicate the main conclusions or interpretations.

Thank you very much for this point.  We fully agree with it and we updated the abstract accordingly.

Show in introduction. What is the purpose of the article’s analysis? The aim of the article needs to be adjusted. Clearly define research questions and answer this questions in the article.  Why is this research important to the reader? What kind of gaps does it cover?

Thank you very much for this point.  We fully agree with it and we updated the last section of the introduction accordingly.

Line 38. Check references style [1,2,3]

Line 56 Check references style [7, 8, 9].

Line 59 Check references style [7, 9, 10, 11]

Line 64 Check references style [4, 12, 13, 14].

Line 66 Check references style [2, 3, 4, 12, 13, 14, 15].

Line 127 Check references style [ 14, 13, 20].

Line 127 Check references Line 315 Check references style [ 1, 33, 32, 42, 43, 44, 40].

Thank you very much for these points. All have been cheeked and corrected.

Literature review. 

The gap or shortcomings of the literature has not been mentioned anywhere in the literature review section. The authors continue to describe different studies but did not mention the gap in the literature. This section must be rewritten as well. It’s really hard to understand what authors mean by mentioning these studies.

Thank you very much for this point. We agree with this point, and we restructured the literature review section. We introduced in the literature review the section of collaborative risks in order to the reader connect the dots between the aim of the proposed strategic process and the gaps in the literature review. We hope that this is it becomes much clearer our objective with the manuscript.

Strategic Process Development and Implementation wrote very long and confusing. It’s really hard to understand what authors want to explain. Add the Materials and Methods section. 

Thank you very much for this point. We fully agree with this point, and we added a Materials and methods section to better explain the implementation and application of the proposed strategic process.

Add source to pictures.

Thank you very much for this point. All Figures illustrated in the document are original except Figure 13 which is an adaptation. The respective reference is now mentioned in the Figure´s description.

The sections describing  the Case study are described in too much detail. 

Thank you very much for this point. We partially agree with this point. The reason is because some other reviewers fully agree with how the case study is illustrated in this manuscript. We also  believe that the case study is the best are to show the potentialities of the proposed strategic process wherefrom readers may get the most valuable and actionable information from.

Show the data volume requirements. How much data needs to be collected for the estimates to be reliable.

Thank you very much for this point. This is a very good question. The proposed strategic process does not have a limitation (upper, or lower) regarding the amount of data that needs to be collected and treated in order to output results. The volume of data that the proposed model analysis is dependent on the available data that an organization has available to be analyzed. This means that the model´s quality outputs is not conditioned by the volume of data available. However, one fact is true: the more data there is available the better is the network-build, and the better are understood and identified behavioral patterns that may have a tendency to head to one of the extremes of the collaborative dimension.

Check references style, depending on the type of work. See the Reference List and Citations Guide for more detailed information.

Thank you very much for this point. All references have been re-checked according to the standards of the journal.

Round 2

Reviewer 1 Report

Much better. Just check a few spelling and formulation errors.

Author Response

Thank you very much once again for your comments. We conducted a third review and corrected a few spelling and formulation errors.

All changes are marked in red.

Reviewer 2 Report

Please, check references style in "References". 

Author Response

Thank you very much once again for your comments. We conducted a third review and corrected a few spelling and formulation errors and rechecked references style in "References". 

All changes are marked in red.